# Inhomogeneous high temperature melting and decoupling of charge density waves in spin-triplet superconductor UTe$_2$

Alexander LaFleur[1,4], Hong Li [1,4], Corey E. Frank[2], Muxian Xu[1], Siyu Cheng [1], Ziqiang Wang [1], Nicholas P. Butch[2,3] & Ilija Zeljkovic [1] ✉

Charge, spin and Cooper-pair density waves have now been widely detected in exotic superconductors. Understanding how these density waves emerge — and become suppressed by external parameters — is a key research direction in condensed matter physics. Here we study the temperature and magnetic-field evolution of charge density waves in the rare spin-triplet superconductor candidate UTe$_2$ using scanning tunneling microscopy/spectroscopy. We reveal that charge modulations composed of three different wave vectors gradually weaken in a spatially inhomogeneous manner, while persisting to surprisingly high temperatures of 10–12 K. We also reveal an unexpected decoupling of the three-component charge density wave state. Our observations match closely to the temperature scale potentially related to short-range magnetic correlations, providing a possible connection between density waves observed by surface probes and intrinsic bulk features. Importantly, charge density wave modulations become suppressed with magnetic field both below and above superconducting $T_c$ in a comparable manner. Our work points towards an intimate connection between hidden magnetic correlations and the origin of the unusual charge density waves in UTe$_2$.

Unconventional superconductivity often occurs in close proximity to other symmetry-breaking phases. In the canonical example of cuprate superconductors, superconductivity is found to co-exist with electronic nematic phase, which breaks the rotation symmetry of the system, and the periodic modulations, or waves, of the charge density and the superfluid density throughout much of the phase diagram[1]. The Cooper pair density waves (PDWs), either as the primary drivers or as secondary states accompanying charge density waves (CDWs), have now been seen in a range of unconventional superconductors, including cuprate high-temperature superconductors[2–4], Fe-based superconductors[5,6] and kagome superconductors[7–9].

Out of the array of density wave superconductors, heavy fermion superconductor UTe$_2$ is an intriguing example of a system that also shows strong evidence for spin-triplet superconductivity in proximity

to an underlying magnetic instability[10,11] and strong fluctuations[12–15]. Unconventional pairing is supported by the very small Knight shift change across the bulk superconducting transition temperature $T_c$[10,16], large and highly anisotropic critical magnetic fields[10], peculiar phase diagram with multiple superconducting regimes[17] and the non-zero polar Kerr effect below $T_c$[12] signaling time-reversal symmetry breaking despite the absence of long-range magnetic order[10]. Within this superconducting state, the system is also reported to host intertwined Cooper pair and charge density wave modulations[18,19], further highlighting the unusual nature of this superconductor. Revealing how these phases emerge as the temperature is lowered, or how they are suppressed by perturbations such as magnetic field or chemical doping can provide an essential insight into their nature. In particular, understanding the CDW transition and how large the CDW onset

[1]Department of Physics, Boston College, Chestnut Hill, MA, USA. [2]NIST Center for Neutron Research, National Institute of Standards and Technology, Gaithersburg, MD, USA. [3]Maryland Quantum Materials Center, Department of Physics, University of Maryland, College Park, MD, USA. [4]These authors contributed equally: Alexander LaFleur, Hong Li. ✉e-mail: ilija.zeljkovic@bc.edu

temperature is, can be crucial for unveiling the nature of the PDW and whether it is a primary PDW.

Here we use spectroscopic imaging scanning tunneling microscopy (SI-STM) to discover an unusual nature of the CDW in UTe$_2$. By investigating the evolution of charge modulations in UTe$_2$ as a function of temperature and magnetic field, we find that the CDW phase emerges substantially above the superconducting transition, at about 10-12 K. SI-STM spatial mapping reveals that the CDW phase is suppressed by forming short-range CDW regions that shrink in size as the CDW approaches global suppression by increasing the temperature. The arrangement of CDW puddles is reproducible with repeated thermal or field cycles, pointing towards the role of local disorder. Interestingly, based on the temperature and energy-dependence, we discover one of the density wave vectors along the mirror symmetry direction is distinct compared to the other two. This reveals an unexpected decoupling of the three-component CDW state. Importantly, the onset temperature of charge modulations observed here closely matches the temperature scale in transport measurements believed to be associated with magnetic correlations, providing a possible connection relating the density waves observed by surface sensitive experiments and bulk measurements. Importantly, we discover that magnetic field gradually suppresses the charge density wave even at temperatures several times higher than the superconducting $T_c$. Such primary CDWs tunable by magnetic field are very rare. Given UTe$_2$ as a spin-triplet superconductor, our work suggests that spin-triplet superconductivity in UTe$_2$ emerges from an unusual charge density wave state connected to "hidden" magnetic correlations as its origin.

## Results

We study bulk single crystals of UTe$_2$ with bulk superconducting $T_{SC} \approx 1.6$ K[10] (Methods). We cleave the crystals in ultra-high-vacuum at liquid nitrogen temperature, and immediately insert them into the microscope head. The crystal structure of UTe$_2$ is orthorhombic and UTe$_2$ crystals tends to cleave along the [0−11] direction[20]. Typical STM topographs of the (0−11) plane show a chain-like surface, with two rows of Te atoms oriented along the [100] direction (Fig. 1a), similarly to what has been observed in other STM studies[18–20]. In addition to the atomic Bragg peaks $Q_{Bragg}$ and Te chain Bragg peaks $Q_{chain}$ (Fig. 1b), Fourier transforms (FTs) of differential conductance d$I$/d$V$ maps show three other pairs of peaks (Fig. 1c, d). These peaks are non-dispersive with energy (Fig. 1e−g) and correspond to an emergent charge density wave (CDW) intertwined with a PDW at the same wave vectors[18,19].

Previous STM experiments primarily focused on studying these density waves in the superconducting state, and reported their existence up to at least 4.2 K[18,19]. Consistent with these, CDW peaks in our data are also clearly detectable at 4.2 K (Fig. 1d). We proceed to

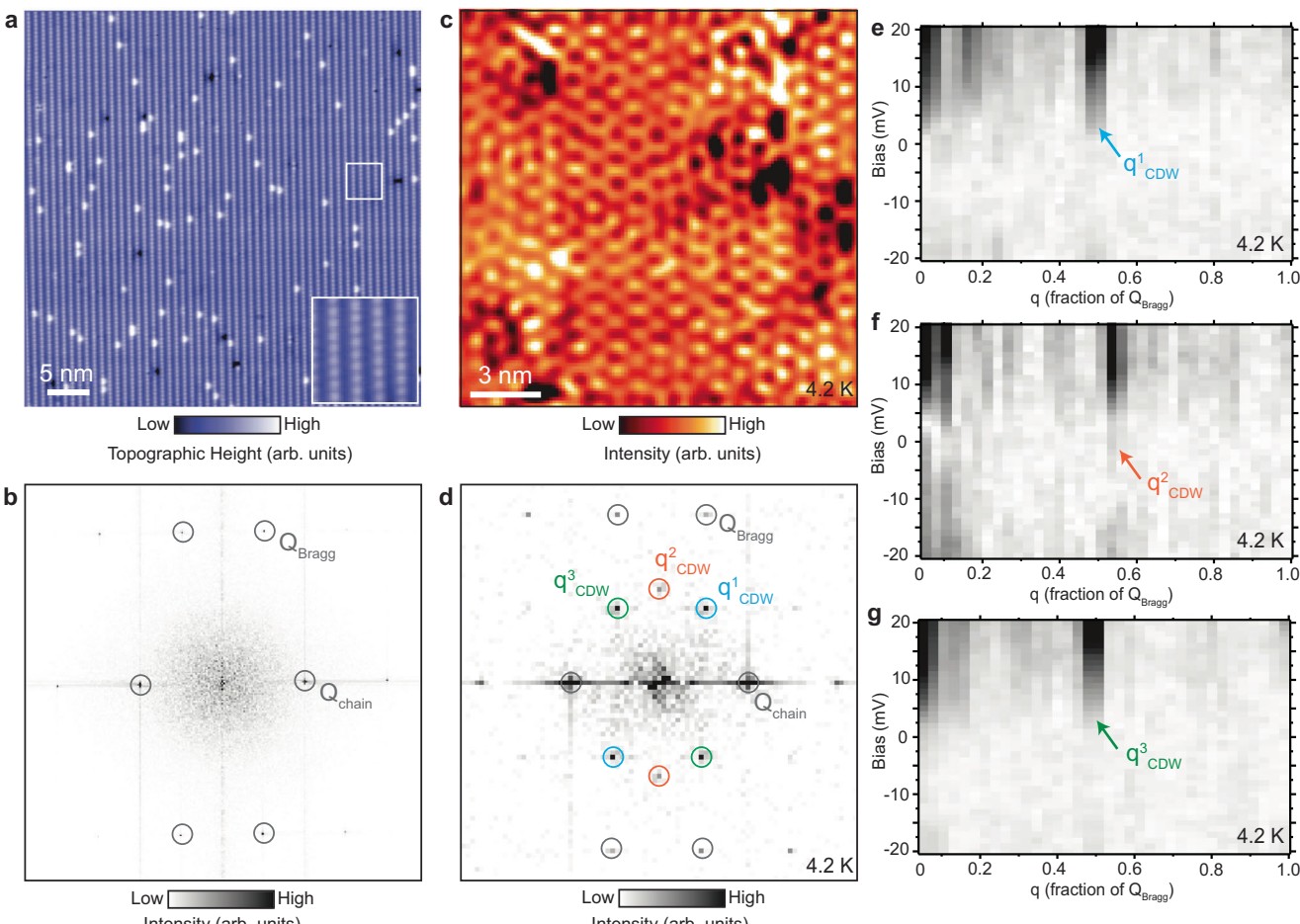

**Fig. 1 | Characterization of the UTe$_2$ surface and the non-dispersive charge modulation peaks. a** Atomically-resolved scanning tunneling microscopy (STM) topograph of UTe$_2$ (0−11) surface with bright rows of Te atoms positioned along the crystal $a$-axis. Inset in (a) shows the zoomed-in topograph over the small white square. **b** Fourier transform (FT) of the STM topograph in (a). Te Bragg peaks ($Q_{Bragg}$) and the unidirectional chain peaks ($Q_{chain}$) are circled. **c** Fourier filtered single-layer d$I$/d$V$ map on the same UTe$_2$ (0−11) surface showing the characteristic charge density wave (CDW) pattern, and (**d**) its associated FT. Three different CDW peaks $q^1_{CDW}$, $q^2_{CDW}$, and $q^3_{CDW}$ are circled in the FT in (**d**) in blue, red and green, respectively. **e−g** FT linecuts, starting from the center of the FT through each of the three CDW peaks, as a function of energy. It can be seen that the position of the three CDW peaks is not dependent on energy. STM setup condition: $V_{sample} = -20$ mV, $I_{set} = 100$ pA (**a**); $V_{sample} = 20$ mV, $I_{set} = 40$ pA, $V_{exc} = 3$ mV (**c**); $V_{sample} = -20$ mV, $I_{set} = 40$ pA, $V_{exc} = 3$ mV (**e−g**).

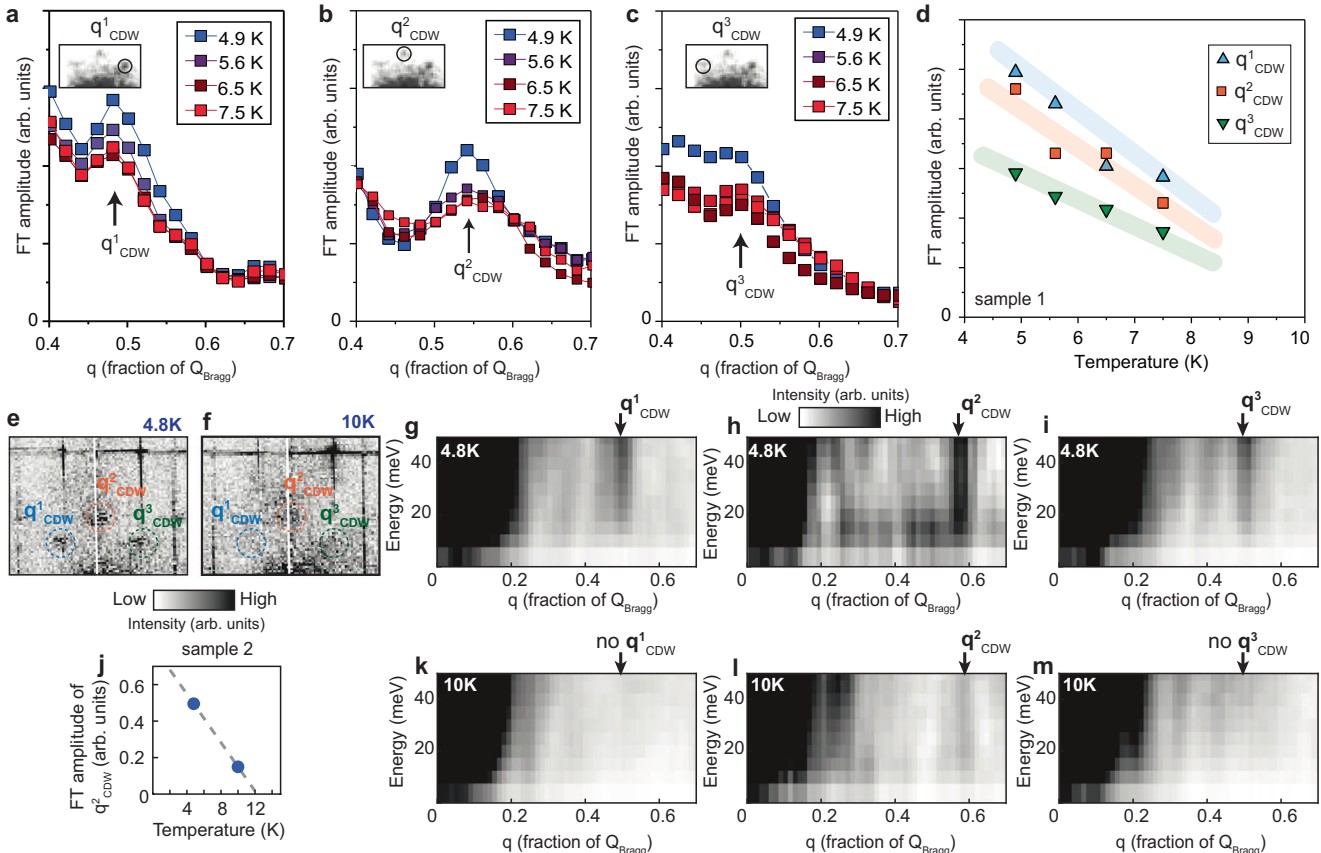

**Fig. 2 | Temperature dependence of the charge density wave modulations.**
**a**–**c** Fourier transform (FT) linecuts of the d$I$/d$V$ map acquired at 40 mV, starting from the center of the FT, through the three CDW peaks: (**a**) $q^1_{CDW}$, (**b**) $q^2_{CDW}$, and (**c**) $q^3_{CDW}$ on sample 1. Suppression of the CDW peaks is apparent with increasing temperature. Insets in (**a**–**c**) show the relevant portion of the FT, with the corresponding wave vector that is plotted circled. **d** Plot of the background-subtracted height (FT amplitude) of the Gaussian fits to each peak in the temperature-dependent linecuts in (**a**–**c**). Blue, red and green lines represent linear visual guides for each of the three CDW peak amplitude dispersions, which seem to approach zero amplitude at about 10 K. **e**, **f** FT of d$I$/d$V$ map acquired at 50 mV on sample 2 at (**e**) 4.8 K and (**f**) 10 K (same scale used for both). The positions of three CDW peaks are circled. **g**–**i** FT linecuts, starting from the center of the FT through each of the three CDW peaks, as a function of energy at 4.8 K. **j** Linear fit of FT peak height of $q^2_{CDW}$ as a function of temperature extracted from (**e**, **f**). **k**–**m** FT linecuts, starting from the center of the FT through each of the three CDW peaks, as a function of energy at 10 K. $q^1_{CDW}$ and $q^3_{CDW}$ cannot be resolved at 10 K, while $q^2_{CDW}$ still shows up with weakened magnitude. Scale bar is the same for (**g**–**i**, **k**–**m**). STM setup condition: $V_{sample} = -50$ mV, $I_{set} = 300$ pA, $V_{exc} = 10$ mV (**e**–**m**).

investigate how the CDW state disappears, first exploring the effect of increasing the temperature. We measure and compare d$I$/d$V$ maps as a function of temperature over an identical area of the sample, acquired under the same experimental conditions (Fig. 2a–d). To quantitatively evaluate the evolution of the CDW phase, we extract FT linecuts from the center of the FT through each of the three CDW peaks at every temperature measured (Fig. 2a–c). It can be seen that peaks get progressively suppressed with temperature. At maximum temperature used in this experiment sequence of 7.4 K, which is about five times higher than the superconducting $T_{SC}$, the peaks are still easily discernable above the background. By plotting the heights of the peaks as a function of temperature, normalized by the FT background signal (Fig. 2d), we estimate the CDW onset temperature $T_{CDW}$ to be about 10 K. This is further confirmed by the data on the second sample, where we can see that CDW peaks are largely suppressed at 10 K, with only $q^2_{CDW}$ still visible (Fig. 2e–m), which we will discuss in more detail in subsequent paragraphs.

Spatial information can provide another important clue on the nature of the CDW suppression. We start by noting that FT of the d$I$/d$V$ maps acquired at about 5 K show diffuse CDW peaks that span several reciprocal-space pixels even after Lawler-Fujita drift-correction algorithm (insets in Fig. 2a–c and e, inset in Fig. 3b), which generally points towards a spatially inhomogeneous order parameter. To visualize this, we Fourier-filter an integrated d$I$/d$V$ map, using a process that removes

all wave vectors except a narrow window around each CDW peak that is exactly 9 pixels in diameter (Fig. 3b). Nanoscale regions with periodic electronic ripples, which correspond to the CDW modulations, and regions where those modulations appear absent can be clearly observed (Fig. 3b), further supporting the notion of an inhomogeneous electronic order approaching the transition.

To gain further insight, we examine Fourier-filtered d$I$/d$V$ maps of each individual CDW peak separately: $q^1_{CDW}$ (Fig. 3d), $q^2_{CDW}$ (Fig. 3e) and $q^3_{CDW}$ (Fig. 3f). In the filtering process, we again select a small FT window that encompasses each individual CDW peak. We note that the overall distribution of regions where CDW is strong vs weak in our data appears comparable regardless of small variations of the filtering window size (more details on the validity of the window size can be found in Supplementary Fig. 1). To quantify its local strength, we plot the intensity of each CDW peak $|q^i_{CDW}|$ (where $i = 1$, 2 or 3) as a function of position to create CDW amplitude maps (Fig. 3g–i). Our first observation is that all amplitude maps show a high level of spatial inhomogeneity, with local regions showing strong modulations dispersed within the larger matrix in which the modulations are substantially weaker. The amplitude maps of the two wave vectors related by the mirror symmetry along the $y$-axis, $q^1_{CDW}$ and $q^3_{CDW}$, exhibit a remarkable similarity with a very high cross-correlation coefficient $\alpha \approx 0.7$ (Fig. 3c). Similar cross-correlation coefficients are observed between maps obtained using

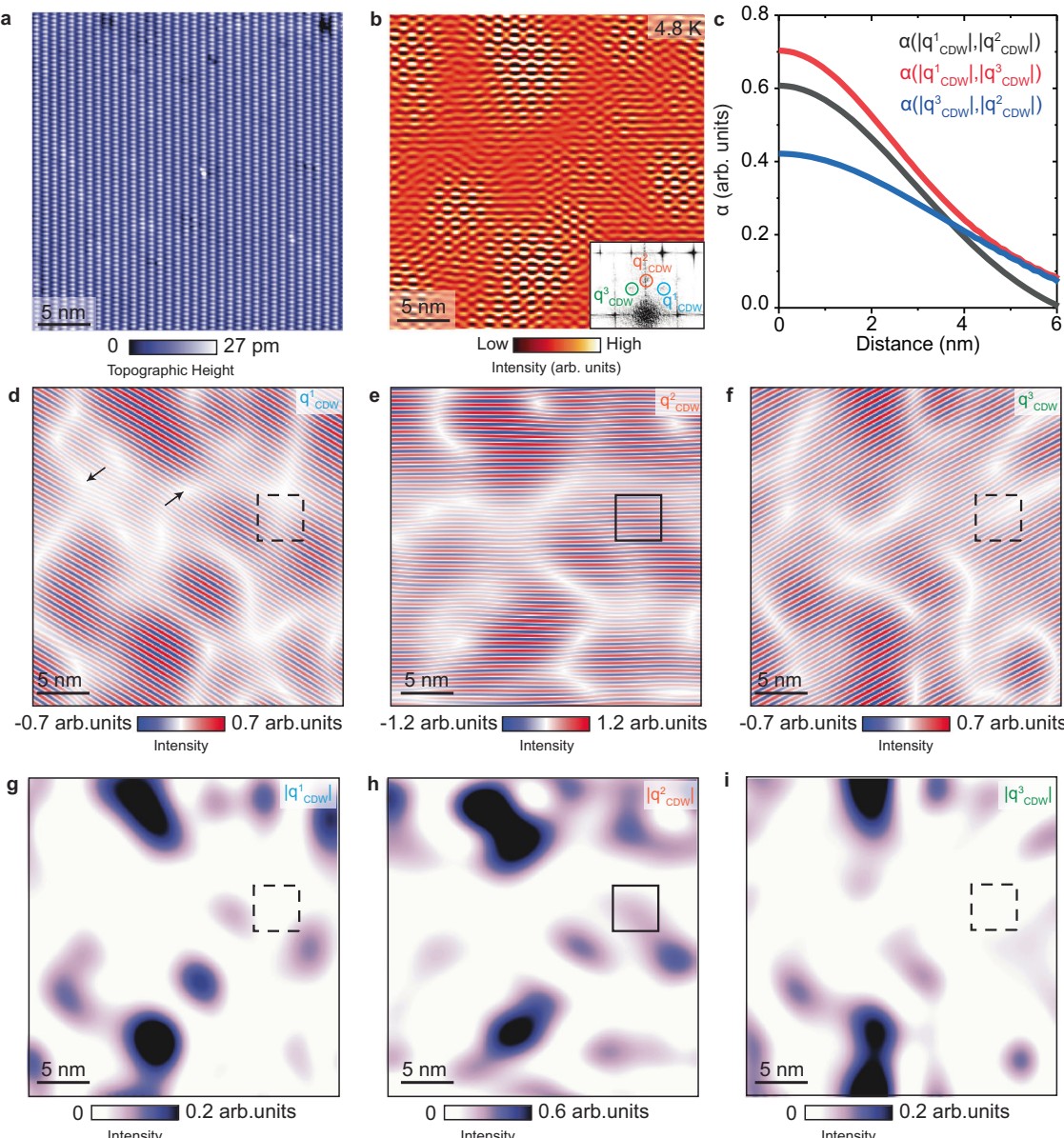

**Fig. 3 | Spatial inhomogeneity and local decoupling of the CDW modulations.** **a** STM topograph and (**b**) filtered integrated d$I$/d$V$ map (adding 4 layers acquired at 35 mV, 40 mV, 45 mV and 50 mV) over the (0−11) surface of UTe$_2$. Fourier filtering is done by removing all wave vectors except 9 pixel diameter centered at each q$^1_{CDW}$, q$^2_{CDW}$ and q$^3_{CDW}$. Inset in (**b**) is the relevant portion of the Fourier transform of the integrated d$I$/d$V$ map, with CDW peaks circled in blue, red and green. **c** Cross-correlation coefficients between the three CDW amplitude maps shown in (**g**–**i**). **d**–**f** Fourier filtered maps extracted from (**a**) by selectively keeping only a narrow

circular FT window centered at (**d**) q$^1_{CDW}$, (**e**) q$^2_{CDW}$ and (**f**) q$^3_{CDW}$. Two arrows in (**d**) point to apparent dislocations that routinely appear in the areas of low CDW amplitude. **g**–**i** CDW amplitude maps extracted from corresponding images in (**d**–**f**), showing the local height of each CDW wave vector. Black squares in (**d**–**i**) mark the same position on the sample with strong (solid line) and weak (dashed line) modulations. STM setup conditions: $V_{sample} = 50$ mV, $I_{set} = 300$ pA, $V_{exc} = 10$ mV.

different sizes of the Fourier filter window (Supplementary Fig. 1). Cross-correlation between the amplitude map of the remaining wave vector, |q$^2_{CDW}$|, and |q$^1_{CDW}$| (or |q$^3_{CDW}$|) is also significant, although somewhat lower (Fig. 3c), suggesting a slightly different morphology of domains. This can be visualized by examining the maps in real space, where we can for example find regions of high |q$^2_{CDW}$|, but low |q$^1_{CDW}$| and |q$^3_{CDW}$| (black squares in Fig. 3d–i). This observation suggests a surprising local decoupling of individual density waves, where q$^2_{CDW}$ becomes decoupled from mirror-symmetric q$^1_{CDW}$ and q$^3_{CDW}$. This decoupling is further supported by examining the FTs of d$I$/d$V$ maps at 10 K – q$^2_{CDW}$ wave vector is still present while the other two, which were prominent at 4.8 K, are now completely suppressed (Fig. 2e–m).

To evaluate the spatial evolution of the CDW order towards $T_{CDW}$, we turn to the temperature dependence of the CDW amplitude maps (Fig. 4). For ease of comparison, we apply the Lawler-Fujita drift correction algorithm[21] to all d$I$/d$V$ maps, which enables us to align data acquired at different temperatures with atomic precision. This process also allows us to systematically apply an identical q-space Fourier filter across all maps examined. For simplicity, we first focus on the q$^1_{CDW}$ peak. Visual comparison of the data sequence from 4.9 K to 6.5 K (Fig. 4a–c) reveals that nanoscale regions where CDW is the most prominent at 4.9 K remain generally the same at the higher temperature. However, amplitude within these nanoscale regions has decreased leading to the appearance that they have shrunk (Fig. 4d–f), consistent with the global suppression of CDW in Fig. 2. Cooling the

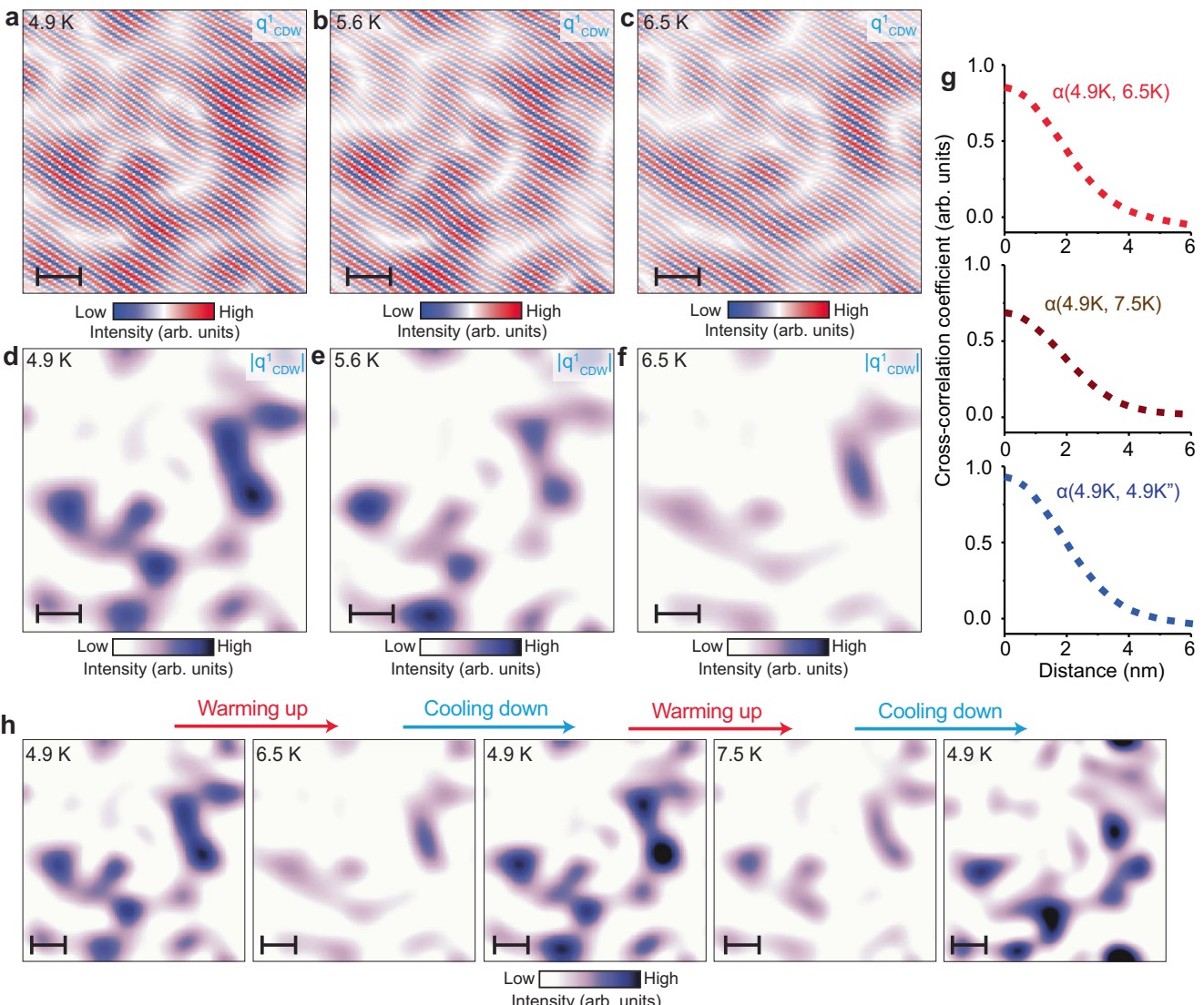

**Fig. 4 | Temperature dependence of the CDW spatial inhomogeneity and robustness to thermal cycles. a–c** Fourier filtered dI/dV maps showing the temperature dependence of the inhomogeneity of $q^1_{CDW}$, and (**d–f**) associated amplitude maps. **g** Radially-averaged cross-correlation coefficient α between pairs of different amplitude maps: α(4.9 K, 6.5 K) (red, top curve), α(4.9 K, 7.5 K) (brown, middle curve), and α(4.9 K, 4.9 K") (blue, bottom curve). Here, 4.9 K" indicates the second cooling to 4.9 K after warming up to 6.5 K. **h** Thermal cycling sequence of amplitude maps associated with $q^1_{CDW}$ taken at the same STM setup conditions. The scale bar in all images is 3 nm. STM setup condition: $V_{sample} = 40$ mV, $V_{exc} = 100$ mV, $I_{set} = 160$ pA.

system back down to the starting temperature of 4.9 K yields the CDW amplitude maps that are virtually indistinguishable from those before the thermal cycle (Fig. 4h). Multiple consecutive thermal cycles paint the same picture of static CDW patches that get suppressed with temperature and expand again upon cooling down (Fig. 4h). The striking similarity of the locations of CDW patches at different temperature is confirmed by the high cross-correlation coefficient (Fig. 4g).

Complementary to temperature dependence, we study if CDW can be suppressed by magnetic field. Previous experiments revealed an intriguing suppression of the CDW in the superconducting state coinciding with the suppression of superconductivity, suggesting an intimate connection between the two[18]. Here we apply magnetic field at 4.8 K, well above the superconducting transition of UTe$_2$. The field is applied at 3 degrees with respect to the direction perpendicular to the sample surface (Fig. 5a, b). We find that the height of CDW peaks slowly decreases with applied magnetic field (Fig. 5c–f). For example, $q^1_{CDW}$ lowers by about 35% from 0 T to 11 T, but remains well above the background (Fig. 5c, e). This suggests a critical field substantially

higher than 11 T. The trend is remarkably consistent with the CDW evolution in the superconducting state[18], although we are in the temperature regime that is three times higher than superconducting $T_{SC}$. To gain spatial information, we extract CDW amplitude maps as a function of magnetic field similarly to our analysis in Figs. 3 and 4. This analysis leads us to two main conclusions. First, CDW patches are again suppressed by the shrinkage of local regions (Fig. 5g, h). Second, CDW puddles revert back to the original morphology after the field is removed (Fig. 5g, h).

## Discussion

Our work reveals how intertwined density waves in a rare spin-triplet superconductor UTe$_2$ get suppressed with temperature, by first forming short-range nanoscale regions with charge modulations embedded within areas where modulations can no longer be observed. Interestingly, we find that the three CDW wave vectors locally decouple. Based on the energy-dependence and the temperature-suppression, we conclude that $q^2_{CDW}$ along the mirror symmetry axis is distinct compared to the other two wave vectors. This in turn suggests a

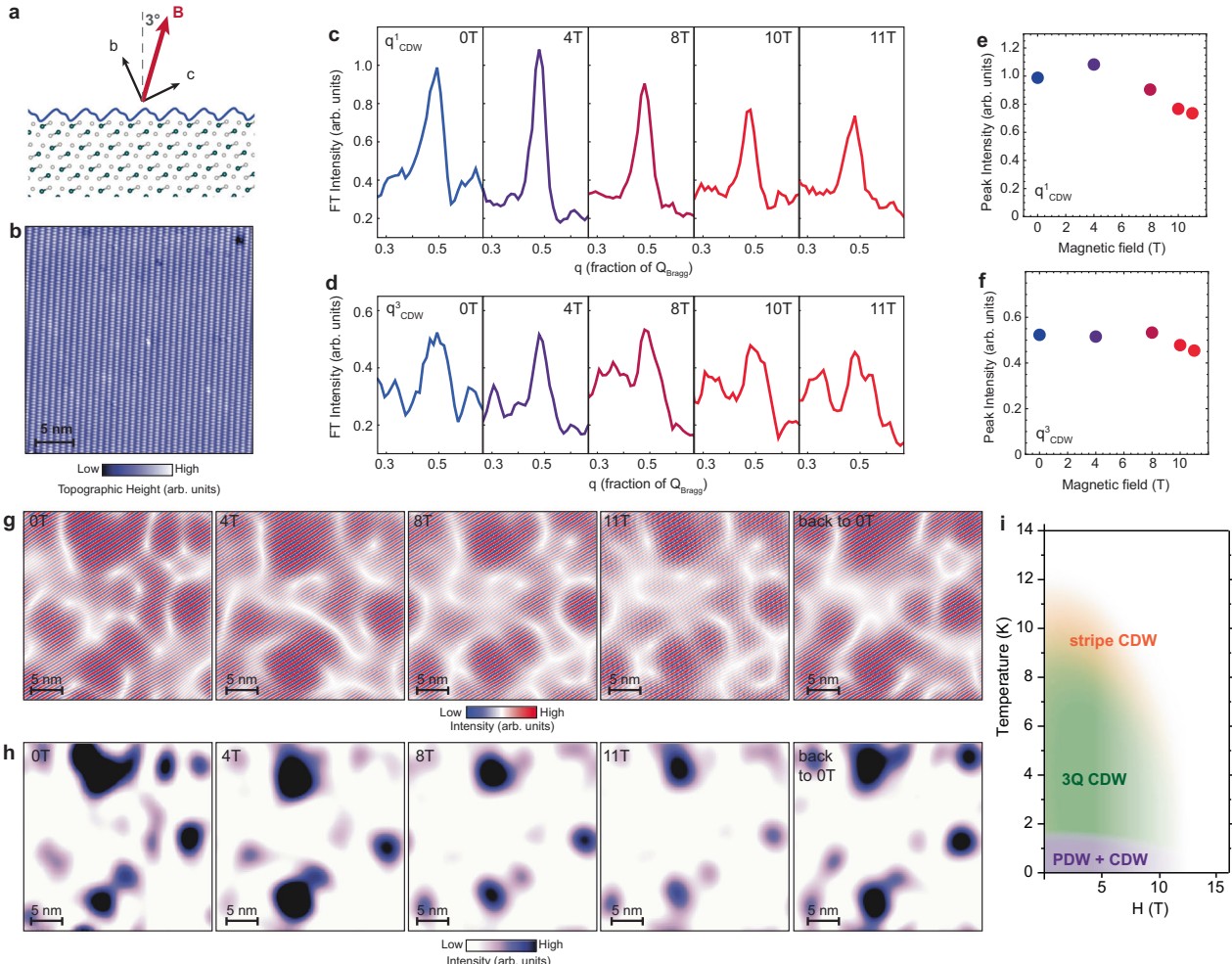

**Fig. 5 | CDW suppression by magnetic field in the normal state of UTe₂ and field sensitivity of spatial inhomogeneity. a** Schematic of the magnetic field direction applied at 3 degrees with respect to the direction perpendicular to the sample surface imaged by STM. This small misalignment is due to accidental sample tilt during the gluing process, and it is determined from the background slope in raw STM topographs. **b** STM topograph of the region of the sample where the data were taken. **c, d** Fourier transform (FT) linecuts from the center of the dI/dV map FT through (**c**) $q^1_{CDW}$ and (**d**) $q^3_{CDW}$ peaks as a function of magnetic field. Sensitivity of the peak heights to large magnetic fields is observed. (**e, f**) Peak heights extracted from (**c, d**) plotted as a function of magnetic field. **g** Fourier filtered dI/dV maps including only the $q^1_{CDW}$ peak (encompassing a 9 pixel diameter around the peak center), and (**h**) associated amplitude maps. **i** Approximate temperature vs magnetic field schematic phase diagram of density waves in UTe₂. Here, "3Q CDW" refers to the charge density wave phase which includes all 3 directions: $q^1_{CDW}$, $q^2_{CDW}$, and $q^3_{CDW}$. "PDW" refers to the pair density wave phase. STM setup condition: $V_{sample} = -50$ mV, $I_{set} = 300$ pA, $V_{exc} = 10$ mV. Data were taken at 4.8 K.

superposition of at least two independent density waves: unidirectional, or "smectic", density wave associated with $q^2_{CDW}$ and a bidirectional combination of mirror-symmetric $q^1_{CDW}$ and $q^3_{CDW}$. It is possible that $q^1_{CDW}$ and $q^3_{CDW}$ would also behave as individual smectic density waves given the different magnetic field dependence reported in ref. 18, although as we discover in Fig. 3, they appear to be spatially intertwined. We note that vortex-like discontinuities seen between regions with strong modulations in Fourier filtered data, also recently reported in ref. 22 (examples of two such features rotated by π with respect to each other are denoted by arrows in Fig. 3d) can dramatically change depending on the size of the filtering window as CDW peaks become less prominent. Moreover, these features routinely appear in the Fourier-filtered data of spatially inhomogeneous orders in other systems (Supplementary Fig. 2). Lastly, as these discontinuities always appear in regions with near-zero amplitude of the order parameter, which is generally below the FT noise threshold, we disregard them in our analysis.

Our experiments also reveal a remarkable repeatability of the morphology of nanoscale CDW regions by magnetic field or temperature cycling. In many materials, atomic-scale crystal imperfections

are often responsible for electronic inhomogeneity[23–25]. Future experiments should explore to what extent accidental impurities or defects could account for the fragmentation or pinning of the charge modulations. This would be especially important to determine given the variation in the properties of nominally stoichiometric crystals of UTe₂, where chemical inhomogeneity could be the natural culprit but defects responsible for this are yet to be identified.

Magnetic field and temperature suppression of the density waves provides narrow constraints on their origin. One possibility could come from a primary PDW. PDWs have now been observed in various materials as periodic modulations of the superconducting gap magnitude and coherence peak heights[2,3,5,7,19]; these density waves naturally get suppressed with temperature as superconductivity is destroyed, possibly leaving a CDW slightly above the superconducting transition. We however find that the CDW persists to a temperature that is almost an order of magnitude higher than the superconducting $T_{SC}$, which would be surprising for a "remnant" CDW phase. The possibility of a primary PDW in UTe₂, which melts into a CDW above $T_{sc}$, was largely supported by the critical magnetic field of the CDW comparable to that of the superconducting critical fields[18]. In our experiments, we find that

similarly high magnetic fields are necessary and can be used to suppress CDW well above $T_{sc}$ (Fig. 5), thus making the immediate connection to superconductivity less obvious. So while a secondary PDW state can still emerge in the superconducting phase, our experiments point towards an unusual primary CDW phase forming well outside the superconducting state.

In general, magnetic field tunable CDWs reported in literature are extremely rare. Prominent examples include field-tunable CDWs in high-temperature superconductor $YBa_2Cu_3O_{7-x}$[26] and moiré graphene[27], but both of these are phenomenologically different from that in $UTe_2$. In $YBa_2Cu_3O_{7-x}$, CDW can only be modified by magnetic field in the superconducting state[26], signaling a direct competition between the two; in contrast in $UTe_2$, CDW is suppressed seemingly unrelated if the material is superconducting or not. In moiré graphene, CDW tunability can be directly tied to the gradual crossing of a Landau level through the Fermi level[27], a physical regime entirely different from $UTe_2$. This points towards a different mechanism of the CDW in $UTe_2$, distinct from the select few other field-tunable CDWs in quantum materials.

Our SI-STM data, which reveal a new 10–12 K temperature scale associated with density waves, provide a fresh clue. Both thermal expansion coefficient data[28] and resistivity measurements[29] reveal features in their data near the same temperature. In particular, a recent pressure study demonstrates that this resistivity feature coincides with the eventual stabilization of the long-range magnetic order; as such, it is attributed to a magnetic energy scale indicative of magnetic fluctuations or a short-range magnetic order[29]. This could provide a plausible link between surface detection of density waves and bulk experiments. This connection, combined with magnetic-field sensitivity of the charge modulations in both superconducting[18] and non-superconducting states (Fig. 5), could point towards the origin of charge modulations in an underlying spin mechanism, without long-range magnetic ordering, which should be tested more directly in future spin-sensitive experiments.

## Methods

Bulk single crystals of $UTe_2$ are grown by the chemical vapor transport method using iodine as the transport agent, with elements of U and Te (atomic ratio 2:3) sealed in an evacuated quartz tube, together with iodine. More details can be found in ref. 10. $UTe_2$ crystals are transported from NIST to BC in a sealed glass vile filled with inert gas to prevent air degradation. The vile is opened in an argon glove box at BC, where the sample is glued to an STM sample holder, and a cleave bar is then attached to the sample using silver epoxy. We transfer the prepared sample from the glove box to an ultra-high vacuum load lock within seconds, and cold-cleave it before putting it into the microscope. STM data were acquired using a customized Unisoku USM1300. Spectroscopic measurements were performed using a standard lock-in technique with 910 Hz frequency. The STM tips used were home-made chemically etched tungsten tips. For the ease of comparison of data acquired at different temperatures and to standardize the Fourier filter windows applied in the analysis, we apply the Lawler–Fujita drift-correction algorithm[21] to our data. This process shifts and aligns the atomic Bragg peaks to be confined to the same individual pixels. Identification of commercial equipment does not imply endorsement by NIST.

## Code availability

The computer code used for data analysis is available upon request from the corresponding authors.

## Data availability

The STM data generated in this study have been deposited in the Zenodo database under accession code 11176886.

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

## Acknowledgements

We thank Johnpierre Paglione for useful discussions. I.Z. gratefully acknowledges the support from DOE Early Career Award DE-SC0020130 for STM measurements. Z.W. acknowledges the support of U.S. Department of Energy, Basic Energy Sciences Grant No. DE-FG02-99ER45747 and the Cottrell SEED Award No. 27856 from Research Corporation for Science Advancement. Sample synthesis and characterization were supported by the National Science Foundation under the Division of Materials Research Grant NSF-DMR 2105191.

## Author contributions

A.L. and H.L. performed the STM experiments and analyzed the data with the help from M.X. and S.C. C.E.F. synthesized and characterized the samples under the supervision of N.P.B. Z.W. provided theoretical input on the interpretation of STM data. A.L, H.L., N.P.B., Z.W. and I.Z. wrote the paper with the input from all the authors. I.Z. supervised the project.

## Competing interests

The authors declare no competing interests.
