## [Peer Review File · Nature Communications]

Inhomogeneous high temperature melting and decoupling of charge density waves in spin-triplet superconductor UTe₂Editorial Note: This manuscript has been previously reviewed at another journal that is not operating a transparent peer review scheme. This document only contains reviewer comments and rebuttal letters for versions considered at Nature Communications.

Reviewers' Comments:

Reviewer #1:

Remarks to the Author:

This paper reports scanning tunneling microscopy (STM) on the temperature- and magnetic-field-dependent charge density wave (CDW) patterns of UTe₂, which are known to be closely related to recently reported pair density waves of superconductivity.

Although peculiar behaviors of CDW in the material have already been reported in previous Nature papers, some new findings, such as magnetic-field-driven suppression at temperatures higher than the critical temperature and q-dependent temperature-driven suppression, are reported in this paper, which might be helpful in revealing peculiar relationships between CDW and pair density waves. Thus, I support the publication of this paper in Nature Communications.

Reviewer #2:

Remarks to the Author:

I have read both the response and the revised manuscript carefully. The authors have thoroughly addressed all issues raised by the three reviewers. Especially, the novelty of this manuscript compared with previous works on CDW/PDW or UTe₂ is highlighted, and the technical issues regarding the FT and decoupling of the three wavevectors are satisfactorily addressed.

This work is a high-quality and timely contribution to the field of unconventional superconductivity. I recommend the publication of the paper in Nature Communications in its present form.